# Synchrotron intensity gradient revealing magnetic fields in galaxy clusters

Yue Hu [1,2] ✉, C. Stuardi [3,4], A. Lazarian [2] ✉, G. Brunetti[4], A. Bonafede[3,4] & Ka Wai Ho[2,5]

Magnetic fields and their dynamical interplay with matter in galaxy clusters contribute to the physical properties and evolution of the intracluster medium. However, the current understanding of the origin and properties of cluster magnetic fields is still limited by observational challenges. In this article, we map the magnetic fields at hundreds-kpc scales of five clusters RXC J1314.4-2515, Abell 2345, Abell 3376, MCXC J0352.4-7401, and El Gordo using the synchrotron intensity gradient technique in conjunction with high-resolution radio observations from the Jansky Very Large Array (JVLA) and the Karoo Array Telescope (MeerKAT). We demonstrate that the magnetic field orientation of radio relics derived from synchrotron intensity gradient is in agreement with that obtained with synchrotron polarization. Most importantly, the synchrotron intensity gradient is not limited by Faraday depolarization in the cluster central regions and allows us to map magnetic fields in the radio halos of RXC J1314.4-2515 and El Gordo. We find that magnetic fields in radio halos exhibit a preferential direction along the major merger axis and show turbulent structures at higher angular resolution. The results are consistent with expectations from numerical simulations, which predict turbulent magnetic fields in cluster mergers that are stirred and amplified by matter motions.

Magnetic fields are a ubiquitous aspect of the cosmos[1–3], and the largest-scale cosmic magnetic fields observed to date are found in galaxy clusters[4–7]. These magnetic fields are a fundamental part of cosmic magneto-genesis, either arising from the turbulent amplification of seed fields or being injected by active galactic nuclei and galaxies[8,9]. They are critical to maintaining energy balance within the intracluster medium (ICM) through heat conduction[10,11], coupling cosmic rays (CRs) to the intracluster gas[12–14], and generating synchrotron radiation by gyrating CR electrons[15]. Despite the crucial importance, the origin of magnetic fields in ICM remains the grand challenge problem. To check the existing theoretical predictions that the

magnetic field can be amplified during galaxy mergers[16–19], a comprehensive understanding of magnetic-field topology in galaxy clusters is imperative.

The study of magnetic fields in galaxy clusters is typically based on observing polarized radio emissions from background or embedded radio galaxies and diffuse synchrotron sources. The Faraday Rotation Measure (RM) of a polarized source, obtained from the rotation of its polarization angle with wavelength, reveals the line-of-sight (LOS) magnetic field weighted by thermal electron density. An RM grid constructed using many background polarized sources provides a valuable probe of the cluster's magnetic-field structure[20,21]. This

[1]Department of Physics, University of Wisconsin-Madison, Madison, WI 53706, USA. [2]Department of Astronomy, University of Wisconsin-Madison, Madison, WI 53706, USA. [3]Dipartimento di Fisica e Astronomia, Universitá di Bologna, Via Gobetti 93/2, I-40129 Bologna, Italy. [4]INAF—Istituto di Radioastronomia di Bologna, Via Gobetti 101, I-40129 Bologna, Italy. [5]Theoretical Division, Los Alamos National Laboratory, Los Alamos, NM 87545, USA. ✉e-mail: yue.hu@wisc.edu; alazarian@facstaff.wisc.edu

approach established that the magnetic fields of the cluster were turbulent and tangled on spatial scales between 5 and 500 kpc[22]. However, these findings are limited by the number of detected background polarized sources, and the uncertainties related to the RM grid interpretation are still not well defined[23]. The plane-of-the-sky (POS) magnetic-field orientation is revealed by direct detection of linearly polarized emission[1,24] but, as we discussed further, this way of probing the magnetic field fails for most of the galactic cluster volume.

Polarized diffuse synchrotron emissions, particularly from radio relics and halos, allow the study of magnetic fields within selected regions of galaxy clusters. For instance, strongly polarized radio relics (with polarization fraction up to 60%) that are typically located at the cluster's periphery can be explored this way[6,19,25]. However, depolarization effects, such as Faraday depolarization caused by thermal electrons and turbulent magnetic fields along the LOS, as well as beam depolarization due to a randomized magnetic-field distribution in the POS, prevent this type of studies for most extended regions of clusters, i.e., radio halos[26,27]. Thus, so far, no polarization mapping of magnetic fields in radio halos has been carried out. This is one of the main challenges faced by the next-generation radio facilities, such as the Square Kilometer Array (SKA)[28,29].

Based on a comprehensive understanding of the pervasive MHD turbulence in the ICM, the refs. [30–32] introduced synchrotron intensity gradient (SIG) as a way to map the magnetic fields. The fast turbulent reconnection in MHD turbulence leads to the mixing of magnetized plasma perpendicular to the direction of the local magnetic field. This results in anisotropic structures of turbulent velocity, magnetic field, and synchrotron radiation elongating along the magnetic field[33,34], as confirmed by numerical simulations[35–38] and in situ measurements in the solar wind[39–41]. As a result, the direction of SIG determines the magnetic field, as was proven by comparing the magnetic fields traced by SIG and synchrotron polarization for the Milky Way magnetic fields[30]. This analysis was further substantiated through numerical simulations that replicated the turbulent conditions within the Milky Way[30]. Nevertheless, it is crucial to recognize that the nature of turbulence within galaxy clusters is distinctly different from the turbulence observed within the Milky Way[42]. Therefore, so far there has been only one attempt to apply SIG for magnetic-field mapping in the relaxed cluster Perseus[32]. The application assumed the Alfvén scale, i.e., the scale at which magnetic fields become dynamically important[43], is observationally resolved. Based on the current study of turbulence and magnetic-field strength in the ICM, the Alfvén scale is constrained from 1 to 60 kpc (see methods, subsection "Synchrotron intensity gradient"). However, our numerical simulations (see methods, subsection "Numerical tests") demonstrated that resolving the Alfvén scale is not essential for SIG. We showed that at scales larger than the Alfvén scale, magnetic fields passively follow the motion of the large-scale flow, with SIG aligned perpendicular to the magnetic field, similar to the case when the Alfvén scale is resolved (see methods, subsection "Synchrotron intensity gradient"). This understanding underlines our confident application of SIG to the ICM.

Here we apply SIG to five disturbed galaxy clusters (RXC J1314.4-2515, Abell 2345, Abell 3376, MXCX J0352.4-7401, and El Gordo), which are of particular interest in terms of magnetic-field amplification and the acceleration of cosmic rays by magnetized turbulence[15,44], using high-resolution radio observations obtained with the Jansky Very Large Array (JVLA)[22,25] and the MeerKAT array[45]. Our choice is motivated by the fact that magnetic fields in Abell 3376's and El Gordo's relics have already been studied by polarization[46,47], while the fields in the relics of RXC J1314.4-2515 and Abell 2345 were also recently mapped[22,25]. Thus, these four clusters provide a valuable test bed for applying SIG. Encouraged by the consistency of the SIG results with observational and numerical testing, we use SIG to map the magnetic fields in the MCXC J0352.4-7401 cluster[45] and we present the magnetic-field measurements in RXC J1314.4-2515 and El Gordo radio halos, revealing the structure of magnetic fields on the largest scales ever measured.

## Results
### Magnetic-field morphologies
In RXC J1314.4-2515, Abell 2345, Abell 3376, MCXC J0352.4-7401, and El Gordo, we follow the SIG[32] adapted to clusters as discussed in methods, subsection "Implementation" to map Plane of Sky (POS) magnetic fields in galaxy clusters RXC J1314.4-2515[25,45], Abell 2345[22], Abell 3376[45], MCXC J0352.4-7401[45], and El Gordo[45]. We define the Alignment Measure (AM) to quantify the alignment of the SIG and the polarization measurements: $AM = 2(\cos^2\theta_r - 1/2)$, where $\theta_r$ is the relative angle between the POS magnetic field inferred from the two methods. $AM = 1$ corresponds to a perfect parallel alignment, while $AM = -1$ represents a perpendicular alignment. The values of the AM are illustrated by chromatically superimposing the values onto the magnetic-field vectors as inferred from polarization measurements and are shown in Figs. 1 and 2, as well as Supplementary Fig. 1.

The detection of polarized synchrotron emission in radio relics in the peripheral regions of RXC J1314.4-2515 and Abell 2345 has been achieved through JVLA observation at a frequency range of 1–2 GHz[22,25]. The resolution of the polarization signal, represented by the full width and half maximum (FWHM) of the Gaussian beam, is around 25″ (or 120 kpc) for RXC J1314.4-2515 and around 30.5″ (or 110 kpc) for Abell 2345. For RXC J1314.4-2515, SIG is calculated per pixel (beam resolution about 7.6″ or 30 kpc) and averaged to FWHM about 25″, which is similar to that of polarization signal. As shown in Figs. 1 and 3, the magnetic fields inferred from SIG and polarization are found to be in agreement (overall AM about 0.70, with a standard deviation of the mean around 0.01), aligned with the elongated intensity relics along the south-north direction. The measured AM is a bit lower compared to its AM obtained for SIG in Milky Way[30]. We attribute this to the higher signal-to-noise and the smaller Faraday rotation effects in the Milky Way case.

SIG and polarization are sensitive to variations of magnetic-field orientation with beam size. As shown in methods, subsection "Numerical tests", AM decreases for a large beam. In addition, unlike SIG, polarization is sensitive to Faraday rotation and Faraday depolarization. As a result, we do expect to see differences between the polarization and SIGs. The systematic difference between the two measures carries important information that sheds light on the difference in the physical mechanisms of the processes that reveal magnetic-field direction, and this difference can be explored in future studies to get deeper insight into the physics of ICM.

Similarly, in Abell 2345, we obtain AM around 0.6 (see Fig. 2). The misalignment between the SIG and polarization in the south tail of Abell 2345-E can be attributed to the potential uncertainties in both measurements (see Supplementary Figs. 2 and 3). In particular, we noticed that the misalignment is associated with a point source. These sources induce synchrotron intensity gradients that are not associated with magnetic fields. The removal of the point source increases the correspondence between polarization and SIG increases. On the other hand, intensity jumps at shock fronts may induce deviation in the SIG from that of the underlying magnetic field. However, as we discuss in the Supplementary Information, section "Uncertainty of the magnetic-field direction measured by the SIG", the contribution of the shock fronts becomes marginal in the process of the sub-block averaging method adopted in SIG. For the relics of Abell 2345, the polarization resolution corresponds to FWHM about 30.5″, which is higher than that of SIG, which potentially also affects the AM. Nevertheless, our study confirms the overall consistency between the directions the polarization and SIG revealed. This finding strengthens the rationale for using this technique in mapping magnetic fields in galaxy clusters where polarization has not been reported.

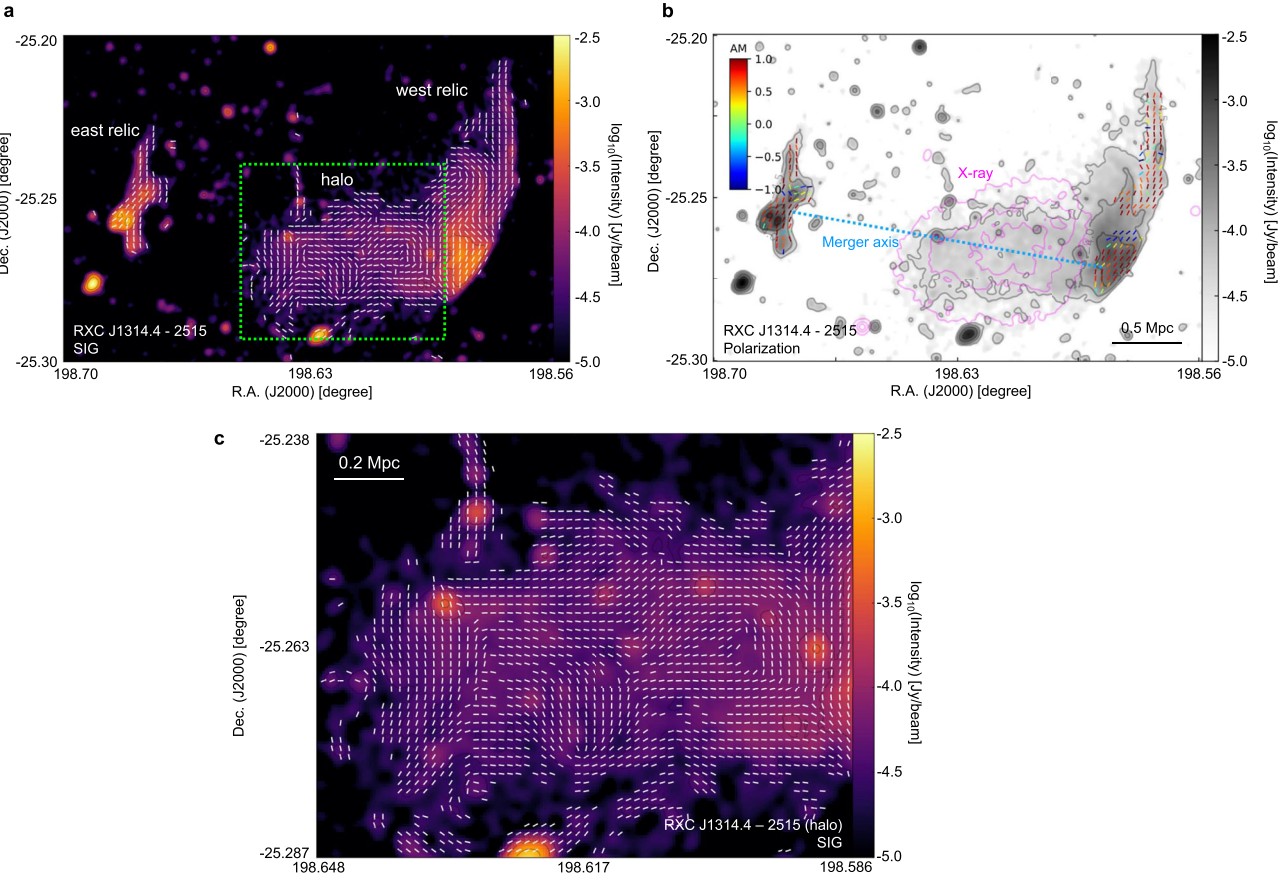

**Fig. 1 | The magnetic-field orientation of the RXC J1314.4-2515 galaxy cluster.**
**a** The morphology of the magnetic fields is revealed through the Synchrotron
Intensity Gradient (SIG, FWHM around 25″ or 120 kpc). **b** The magnetic-field mor-
phology is revealed through JVLA synchrotron polarization at 3 GHz (FWHM
approximately 25″). Each magnetic-field segment represents the SIG (or polariza-
tion) averaged for 6 × 6 pixels for visualization purposes. The colors of the polar-
ization segments represent the AM of the SIG and polarization. The magnetic field is
overlaid on the higher resolution synchrotron emission image from the MeerKAT
survey[45] at 1.28 GHz (FWHM arond 7.6″ or 30 kpc). The pink contours represent
X-ray emission measured by the XMM-Newton and the dotted line indicates the
expected merger axis determined by radio images. The merger axis defined by
X-ray emission's elongation is studied in the Supplementary Figs. 4 and 5. **c** A zoom-
in view of the magnetic field in RXC J1314.4-2515's halo, indicated by the green box
in the top panel, is provided. Source data are provided as a Source Data file.

Figures 4, 5, and 6 present the magnetic-field measurement of
the merging clusters Abell 3376, MXCX J0352.4-7401, and El Gordo
using the SIG. These clusters have different redshifts (*z* approxi-
mately 0.87 for El Gordo, 0.127 for MXCX J0352.4-7401, and 0.046
for Abell 3376). The SIG measurements are in agreement with
earlier partial polarization observations in Abell 3376's double
relics[46] and El Gordo's west relic[47]. The magnetic fields associated
with radio galaxy jets and bent by ICM in Abell 3376 have also been
observed in the SIG measurement[48]. These earlier studies only
covered limited portions of the relics with detected polarization
signals, but the SIG measurements cover the entire structure and
provide complete magnetic-field maps. In addition, we present
the SIG-mapped magnetic field for the detected relics in MCXC
J0352.4-7401. Future high-resolution synchrotron intensity
observations will enable SIG to map magnetic fields at smaller
scales.

Measurement of polarized synchrotron emission in radio halos is
challenging due to the strong Faraday depolarization effect. SIG,
however, offers a unique solution to this problem as it is insensitive to
depolarization. As a result, SIG opens an avenue for studying the
magnetic field of radio halos. For example, in RXC J1314.4-2515 and El
Gordo, polarization is detected only in the double relics[25] and the west
relic[47], respectively. However, SIG provides a complete picture of the
magnetic fields in the entire cluster, including halos (see Figs. 1 and 6).

Due to the beam and LOS averaging, the gradient signals caused by
small-scale fluctuations are averaged out in the central radio halo. SIG,
however, remains sensitive to the large-scale component of the mag-
netic field (see methods, subsection "Synchrotron intensity gradient").
At low resolution, the magnetic field in RXC J1314.4-2515 (see Supple-
mentary Fig. 1) is preferentially aligned along the merger axis, whereas
on top of this behavior, vortex-like start appearing at higher resolution
(see Fig. 1), yet limited to effective 120 kpc resolution. The maps clearly
show the transition between radio relics and halos, which are asso-
ciated with shock waves where the magnetic field is compressed in the
direction perpendicular to the merger axis (see Supplementary Figs. 4
and 5).

## Discussion

The reported magnetic-field structure testifies to the magnetic-field
amplification during galactic mergers. It is in line with previous RM
grid measurements[22] and MHD simulations that predicted magnetic
fields evolve with cluster dynamics[16–19] (see Fig. 7 for an illustration).
The fields are stretched/stirred and further amplified by large-scale
bulk flows along the merger axis. On the other hand, the presence of
large-scale magnetic fields suggests that the magnetic-field ampli-
fication through turbulent dynamo is rather inefficient at its non-
linear stage, in which less than 10% of turbulent kinetic energy is
transferred to magnetic-field energy[44,49]. Thus, for stationary

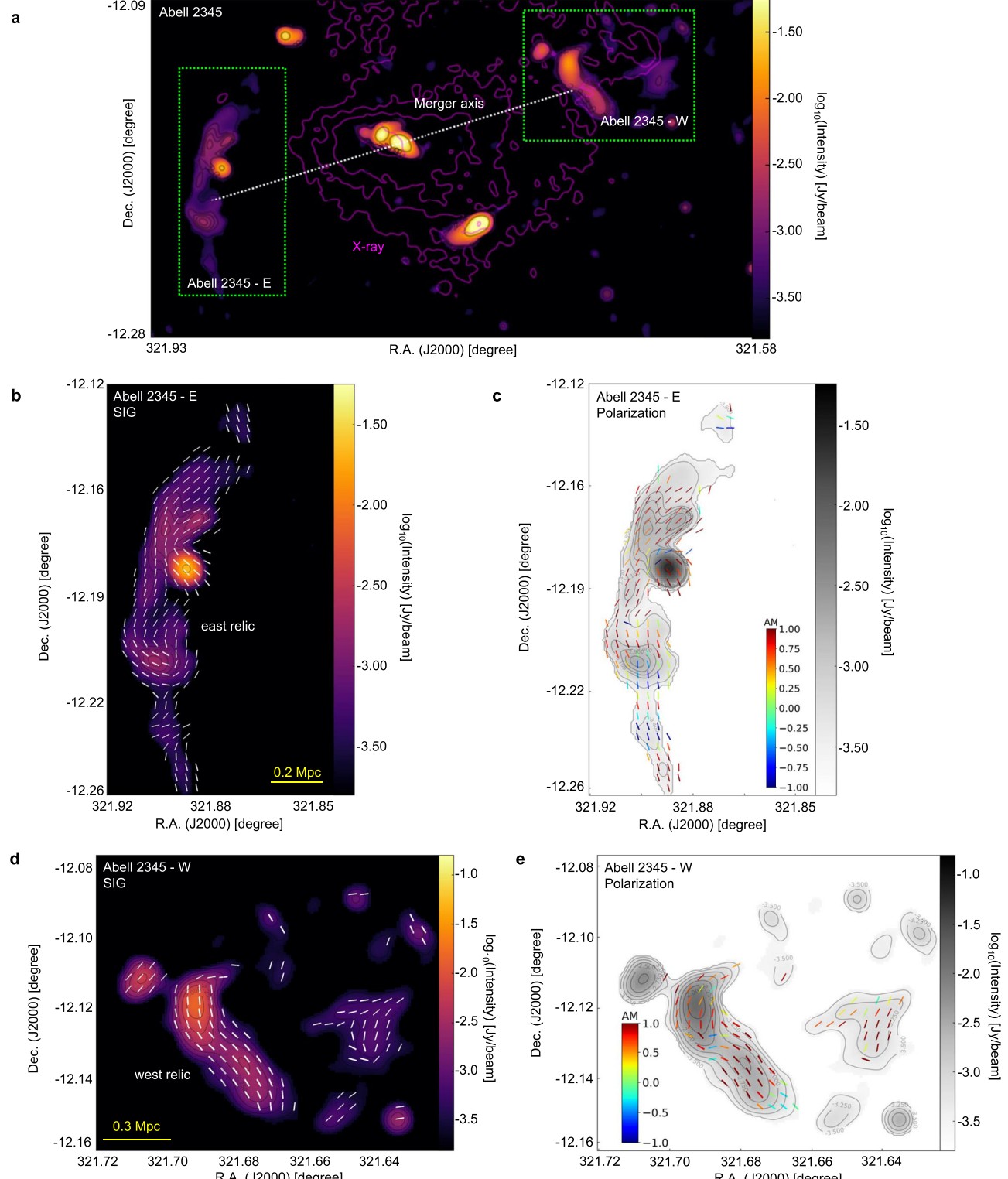

**Fig. 2 | The magnetic-field orientation of the Abell 2345 galaxy cluster.**
**a** Synchrotron emission image of the Abell 2345 cluster observed with the JVLA at 1.5 GHz. The pink contours represent X-ray emission measured by the XMM-Newton and the dotted line indicates the expected merger axis. **b**, **c** Same as Fig. 1a, b, respectively, but for the Abell 2345 cluster's subregions E (**b**) and W (**c**), indicated by the green boxes in the **a**. The magnetic field inferred from the synchrotron polarization and background emission image is from the JVLA observation at 1.5 GHz (FWHM approximately 30.5″ or 110 kpc). The resolution of Synchrotron Intensity Gradient (SIG) measurement is approximately 50″ (or 180 kpc). **d**, **e** same as panes **b**, **c**, respectively, but for the Abell 2345 cluster's subregion W. Source data are provided as a Source Data file.

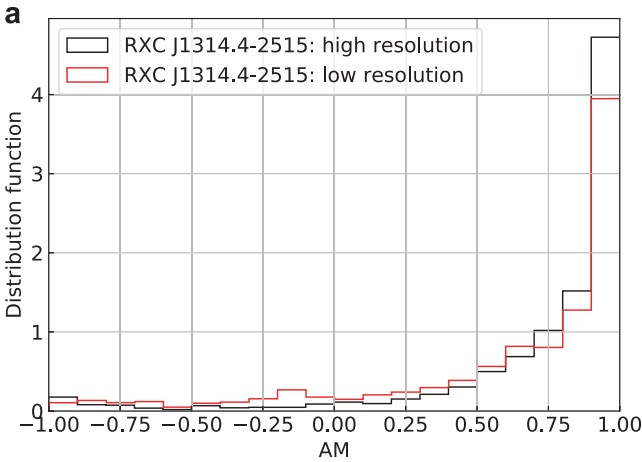
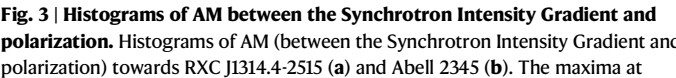
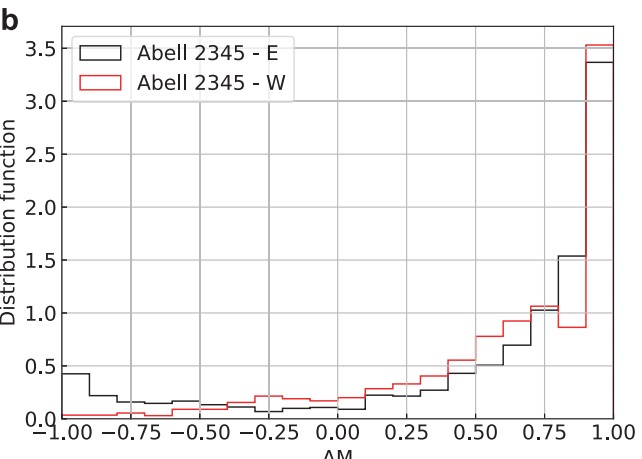

**Fig. 3 | Histograms of AM between the Synchrotron Intensity Gradient and polarization.** Histograms of AM (between the Synchrotron Intensity Gradient and polarization) towards RXC J1314.4-2515 (**a**) and Abell 2345 (**b**). The maxima at

AM = 1 suggests an excellent alignment of magnetic fields revealed by the two methods. Source data are provided as a Source Data file.

turbulence, it would take around ten of the longest eddy turnover times for the magnetic field to reach equipartition with turbulence. Since the merging of clusters did not last for such a long time, the more prominent magnetic-field structures are formed through the large-scale stretching caused by the flows resulting from the merger. We anticipate that there are more magnetic field stochasticity on the small scale.

Our results open perspectives to map magnetic fields in clusters and large-scale structures and allow for the comparison between numerical expectations of merging clusters and observations. SIG's ability to trace the galaxy clusters' magnetic fields is confirmed by (a) MHD simulations presented in Fig. 8, (b) the correspondence of SIG with polarization measurements in radio relics (see Figs. 4 and 5), as well as (c) the correspondence of the SIG-traced halo magnetic-field structure to those numerically predicted in the refs. 16–19. Being insensitive to the Faraday depolarization, SIG can be applied to many clusters with diffuse radio emission, which is especially timely in view of the coming SKA and the Low-Frequency Array (LOFAR) observations. It opens a unique way of using radio data for the regions where depolarization masks and distorts the polarized signal. The prospects of SIG get more exciting in view of recent LOFAR observations that discovered synchrotron radiation on large scales (several Mpc), on the outskirts of the cluster, or between massive cluster pairs[50–52]. As soon as high signal-to-noise images of this very large-scale synchrotron emission become available, SIG will also enable us to map these largest-scale magnetic fields and study their statistic properties (see Supplementary Fig. 6 for an example of the statistical analysis via the structure-function). This will provide constraints on the theories of magneto-genesis of magnetic field and their role within large-scale structures of Universe evolution.

## Methods
### Synchrotron intensity gradient
Synchrotron intensity gradient (SIG) introduced in refs. 30–32 is extended in this paper for super-Alfvénic turbulence present in the ICM. In the super-Alfvénic regime (i.e., $M_A > 1$), turbulent motions at the injection scale are hydrodynamic, and the kinetic energy of the turbulence follows an isotropic Kolmogorov cascade ($v_l \approx l^{1/3}$, where $v_l$ is the velocity of the turbulence at scale $l$). However, at smaller scales, the backreaction of the magnetic field becomes stronger and the turbulence becomes anisotropic on the Alfvén scale $l_A = LM_A^{-3}$, where $L$ is the injection scale[53]. The typical Alfvén

scale in galaxy clusters can be calculated as follows[43]:

$$l_A \approx 100 \left(\frac{B}{\mu G}\right)^3 \left(\frac{L}{300\,\text{kpc}}\right) \left(\frac{v_L}{10^3\,\text{kms}^{-1}}\right)^{-3} \left(\frac{n_e}{10^{-3}\,\text{cm}^{-3}}\right)^{-\frac{3}{2}}\,\text{pc,} \quad (1)$$

where $B$ is the magnetic-field strength, $L$ is the injection scale of turbulence, and $n_e$ is the electron number density. Based on typical values of $B$ (0.5–2.0 µG), $L$ (300–600 kpc), $v_L$ (100–300 km s⁻¹), and $n_e$ (10⁻³ cm⁻³) from literature[25,54–59], the Alfvén scale $l_A$ ranges from approximately 1–60 kpc. Note that quoted values of the physical quantities are typically measured for cluster cores. In the edges of the cluster, these values may be different.

At scales smaller than $l_A$, the anisotropic MHD turbulence causes the turbulent eddies to elongate along the magnetic field, resulting in a gradient perpendicular to the field. At scales larger than $l_A$, the large-scale gas flows can still regulate the dynamically unimportant magnetic field in the ICM, causing the field to follow or elongate along large-scale structures, as shown in Fig. 8. Therefore, the actual number of $l_A$ is not important for using SIG to map the magnetic field, because the gradients are perpendicular to the magnetic field in both cases.

### Comparison with polarization
SIG and synchrotron polarization are based on different physical effects to reveal the magnetic field. While synchrotron polarization emerges from the magnetic fields' effects on relativistic electrons, SIG is grounded in the interaction between magnetic fields and conducting fluid. Notably, measurements obtained from both methods are subject to the effects of LOS averaging. Given the expected scenario where the LOS integration length for cluster halos surpasses the scale of magnetic-field entanglement, turbulence-induced fluctuations are not anticipated to manifest a preferential direction, instead accumulating through a process akin to a random walk. These turbulence-driven fluctuations play a pivotal role in decreasing the synchrotron polarization fraction, known as Faraday depolarization. However, SIG is immune to the depolarization effect, maintaining its reliability in tracing the magnetic fields in super-Alfénic radio halos. As illustrated in Fig. 8, the alignment between SIG and magnetic fields remains statistically stable, even when the LOS integration length increases. On the other side, polarization is sensitive not only to the value of the magnetic-field strength that aligns parallel to the LOS but also to the density of thermal electrons within the environment. Given that the orientation of magnetic fields within galaxy clusters is subject to

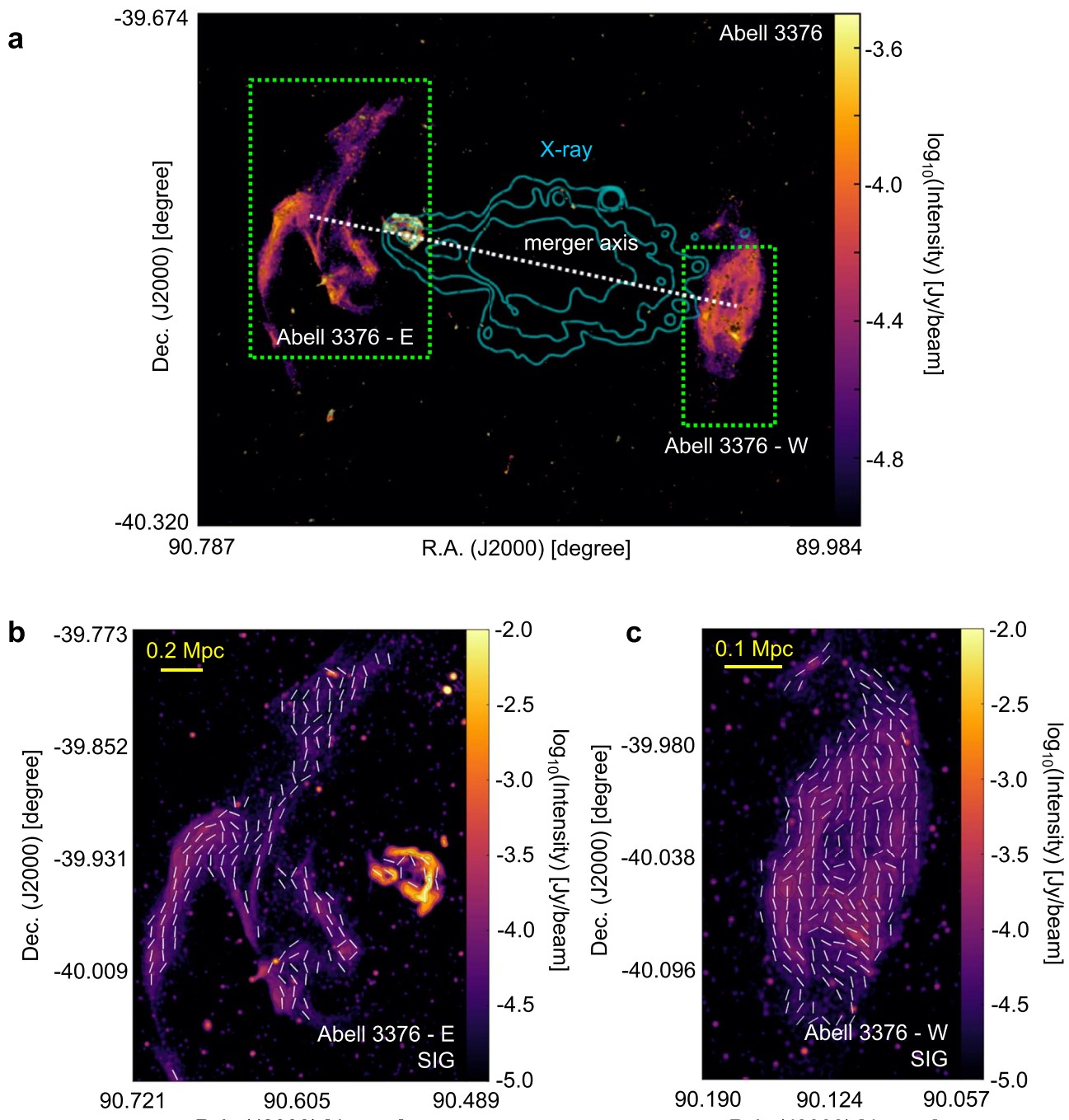

**Fig. 4 | The magnetic-field orientation of the Abell 3376 galaxy cluster. a** Overall synchrotron emission intensity map from MeerKAT observation at 1.28 GHz (FWHM approximately 7″) of the Abell 3376 cluster. The cyan contours represent X-ray emission obtained from the XMM-Newton archival observations[65] and the dotted line indicates the expected merger axis. **b, c** The magnetic-field orientation (FWHM approximately 24″ corresponding to a physical scale of 22 kpc), represented by white segments, in Abell 3376's E (**b**) and W (**c**) relics (indicated by the green boxes in **a**). Source data are provided as a Source Data file.

changes along the LOS, there is a possibility for differences between contributions to SIG and those to polarization.

A factor that can induce misalignment in observational applications is the beam size. This effect can be illustrated by considering a particular example of a very large beam, which covers the full field of view of the observation. Such a beam results in a constant intensity map and the corresponding gradient vanishes, but polarization still can give one magnetic-field orientation for this large beam. This beam effect, therefore, reduces the alignment as shown in the right panel of Fig. 8.

## Implementation

In this study, SIG serves as the primary tool for analysis. The intensity gradient calculated from the synchrotron intensity map ($I(x, y)$) allows for mapping the orientation of magnetic fields. This is achieved through a pixelized gradient map $\psi(x, y)$ as follows:

$$\nabla_x I(x, y) = I(x + 1, y) - I(x, y), \quad (2)$$

$$\nabla_y I(x, y) = I(x, y + 1) - I(x, y), \quad (3)$$

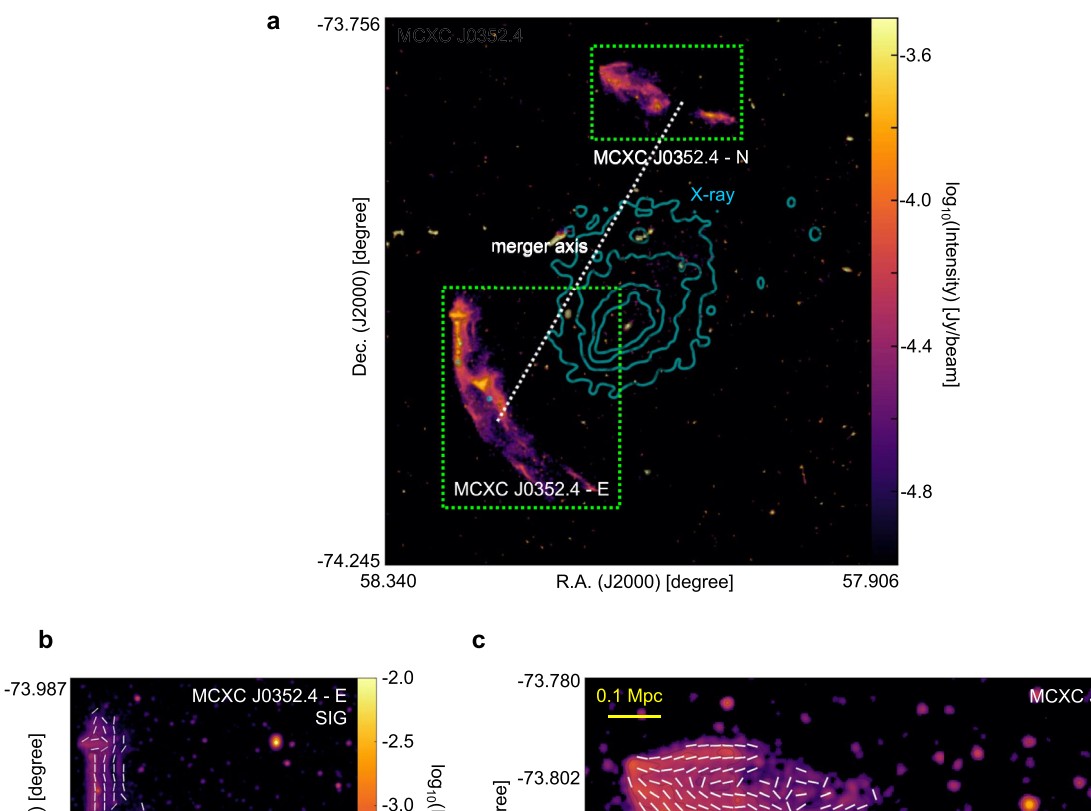

**Fig. 5 | The magnetic-field orientation of the MCXC J0352.4-7401 cluster.**
**a** Overall synchrotron emission intensity map from MeerKAT observation at 1.28 GHz (FWHM approximately 7″) of the MCXC J0352.4-7401 clusters. The cyan contours represent X-ray emission obtained from the XMM-Newton archival observations[66]

and the dotted line indicates the expected merger axis. **b**, **c** The magnetic-field orientation (FWHM approximately 24″ corresponding to a physical scale of 62 kpc), represented by white segments, in MCXC J0352.4-7401's E (**b**) and N (**c**) relics (indicated by the green boxes in **a**). Source data are provided as a Source Data file.

$$\psi(x, y) = \tan^{-1}\left(\frac{\nabla_y I(x, y)}{\nabla_x I(x, y)}\right), \qquad (4)$$

here, $\nabla_x I(x, y)$ and $\nabla_y I(x, y)$ represent the $x$ and $y$ components of the gradient, respectively. Gradients are blanked out if their corresponding intensity value is less than $3\sigma$ noise level.

The gradient map $\psi(x, y)$ is further processed through the sub-block averaging method[60]. This method involves taking all gradient orientations within a sub-block of interest and applying Gaussian fitting to the corresponding histogram. The peak value of the Gaussian distribution represents the statistically most probable gradient orientation within that sub-block. The averaging step ensures that the resulting gradient direction incorporates turbulence's statistical properties. The processed gradient map is denoted as $\psi_s(x, y)$, and its uncertainty is related to the sub-block size. A larger sub-block size guarantees a sufficient amount of data for statistical fitting, leading to lower uncertainty. Typically, gradients are averaged over $20 \times 20$-pixel sub-blocks, as this size has been determined through previous numerical and observational

studies to guarantee sufficient statistics for extracting turbulence's properties[61]. To address the boundary effect in cases where the number of data points at the edge of an intensity structure may be less than $20 \times 20$ pixels, a minimum of $10 \times 10$ pixels is established for averaging.

The averaging procedure for each sub-block is independent. However, this is not the case for actual magnetic-field lines, necessitating a correlation of the averaged gradient with that of its neighboring. This can be mathematically handled by performing smoothing on the pseudo-Stokes parameters ($Q_g$ and $U_g$), which are defined as:

$$Q_g(x, y) = I(x, y)\cos(2\psi_s(x, y)), \qquad (5)$$

$$U_g(x, y) = I(x, y)\sin(2\psi_s(x, y)), \qquad (6)$$

$$\psi_g(x, y) = \frac{1}{2}\tan^{-1}\left(\frac{U_g}{Q_g}\right), \qquad (7)$$

where $\psi_g$ is the pseudo polarization angle. Similar to the synchrotron polarization, $\psi_B = \psi_g + \pi/2$ gives the POS magnetic-field orientation. The weighted intensity ensures that (i) $Q_g$ and $U_g$ follow a Gaussian distribution, which facilitates the smoothing of the pseudo-Stokes parameters using a Gaussian filter. The FWHM of the Gaussian filter is

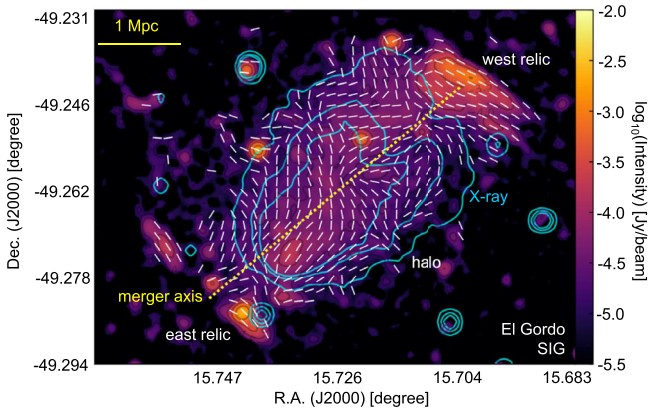

**Fig. 6 | The magnetic-field orientation of the El Gordo cluster.** The background emission image is from MeerKAT observation at 1.28 GHz (FWHM approximately 7″). The Synchrotron Intensity Gradient (SIG) measurement has a resolution of FWHM approximately 24″ (or 400 kpc). Each white segment represents the SIG averaged for 6 × 6 pixels for visualization purposes. The blue contours represent the X-ray emission obtained from the Chandra archival observations[67] and the dotted line indicates the expected merger axis. Source data are provided as a Source Data file.

equal to the sub-block size. (ii) The magnetic fields mapped by SIG are intensity-weighted, which is also the case for the magnetic field inferred from synchrotron polarization weighted by the polarized intensity.

## Uncertainties

The uncertainty in SIG is mainly due to systematic errors in radio images and the SIG algorithm itself. We calculated SIG's uncertainty by considering error propagation and blanked out the pixels in which the uncertainty is larger than 30 degrees. We presented the uncertainty maps in Supplementary Figs. 2 and 3 and listed the median value of the uncertainty for each cluster in Table 2.

## Numerical tests

A numerical test is presented for illustration purposes. In accordance with the method described in the ref. 30, we utilize several 3D MHD simulations of turbulence to synthesize synchrotron emission maps, including the synchrotron intensity $I_s(x, y)$, Stokes parameters $Q_s(x, y)$ and $U_s(x, y)$, and polarization angle $\psi(x, y)$. Each simulation box is divided into $512^3$ cells, with a turbulence injection scale of approximately 256 cells. The calculation of these maps is based on the following equations:

$$I_s = \int n_{e,r}(B_x^2 + B_y^2)B_\perp^\gamma \, dz, \tag{8}$$

$$Q_s = \int -n_{e,r}(B_x^2 - B_y^2)B_\perp^\gamma \, dz, \tag{9}$$

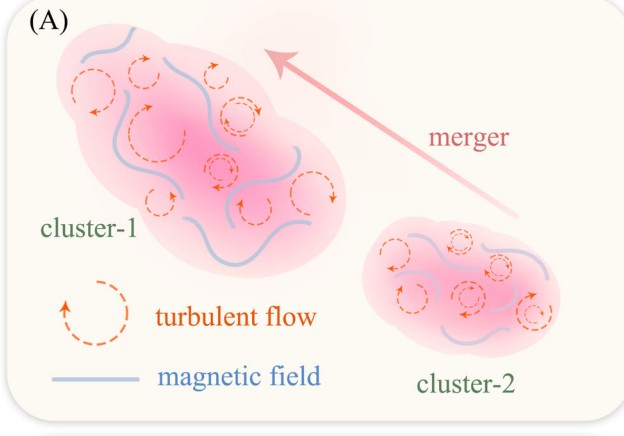
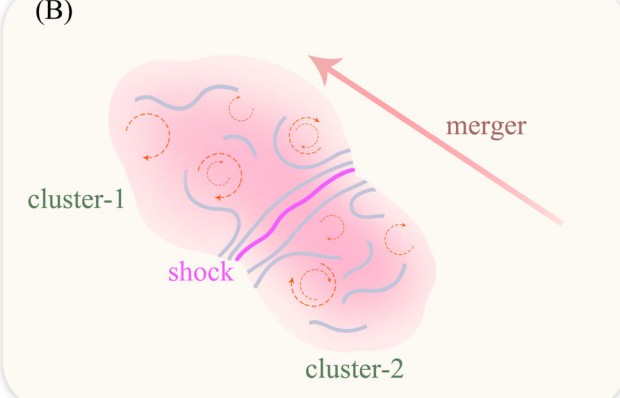
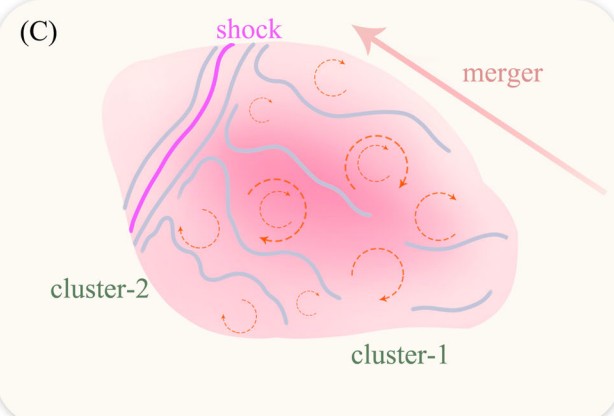
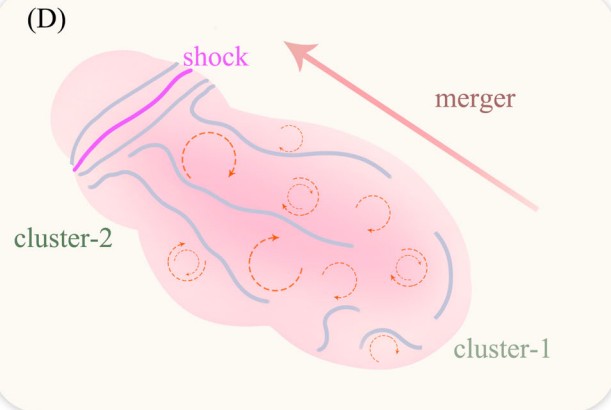

**Fig. 7 | Cartoon illustration of magnetic-field evolution in cluster merger.** In the merging of two turbulent clusters (**A**): cluster-1 and cluster-2, the magnetic field is draped and amplified at the merger (advancing) shock in the first phase (**B**), and then the field is stretched along the merger axis (**C**), and finally it is further amplified by turbulence generated in the cluster merger (**D**).

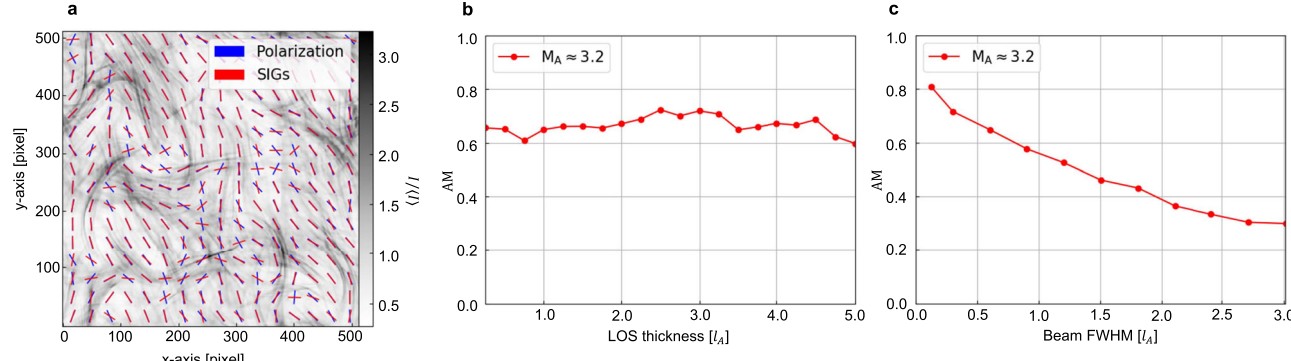

**Fig. 8 | Numerical test of SIG. a** Comparison of the magnetic-field orientation inferred from Synchrotron Intensity Gradient (SIG, red segment) and polarization (blue segment). **b**, **c** AM of magnetic fields inferred from SIG and polarization as a function of the cluster's LOS thickness (**b**) and the beam size (**c**). Polarization is smoothed to match the resolution of SIG after averaging. Source data are provided as a Source Data file.

$$U_s = \int -n_{e,r}(2B_xB_y)B_\perp^y \, dz, \tag{10}$$

$$\psi = \frac{1}{2}\tan^{-1}\left(\frac{U_s}{Q_s}\right), \tag{11}$$

where $B_\perp = \sqrt{B_x^2 + B_y^2}$ is the magnetic-field component perpendicular to the LOS, with $B_x$ and $B_y$ being the $x$ and $y$ components, respectively, and $n_{e,r}$ is the density of relativistic electrons. In consideration of the fact that the anisotropy of synchrotron emission is insensitive to the spectral index of the electron energy distribution[62], a homogeneous and isotropic distribution with spectral index $\alpha = 3$ is adopted, yielding an index of $\gamma = (\alpha - 3)/4$.

An example is shown in the left panel of Fig. 8. The super-Alfvénic condition of $M_A \approx 3.2$ leads to a turbulence length scale of $l_A \approx 8$ cells. The magnetic-field orientation inferred from SIG globally agrees with that derived from polarization (with an agreement measure of AM ≈ 0.81). The AM values for other simulations are listed in Table 1, which demonstrates the correspondence of the magnetic-field orientation mapped by polarization and SIG.

### Observations
The synchrotron emission images used in this work are produced with JVLA (https://data.nrao.edu/portal/#/) and MeerKAT observations (https://archive-gw-1.kat.ac.za/public/repository/10.48479/7epd-w356/index.html). The summary of the dataset is presented in Table 2.

For RXC J1314.4-2515, polarization images with similar resolution thanks to the images released by the MeerKAT Galaxy Cluster Legacy Survey Data Release 1. The high-resolution 1.28 GHz image from which we derived the SIG measurements (see Fig. 1) has a beam size of 7.3″ × 7.6″ corresponding to a spatial resolution of about 30 kpc at $z = 0.247$. The RMS noise of this image is 5 μJy/beam. Instead, a low-resolution SIG image was obtained from the same JVLA dataset from which we derived polarization (see Supplementary Fig. 1). This total intensity image at 3 GHz has a resolution of 17″ × 17″, corresponding to 66 kpc around, and an RMS noise of 13 μJy/beam. This image is the

same as the one presented by the ref. 25 but smoothed with a circular beam size. As for Abell 2345, the resolution beam of the 1.5 GHz JVLA total intensity map in Fig. 2 is 19″ × 19″, corresponding to a spatial resolution of 58 kpc at $z = 0.179$. With respect to the image presented by the ref. 22, this has been smoothed to obtain a circular beam size. The RMS noise is 80 μJy/beam. In terms of Abell 3376, the resolution beam of the 1.28 GHz MeerKAT total intensity images of this cluster shown in Fig. 4 is 7.4″ × 7.6″, corresponding to a spatial resolution of approximately 7 kpc at $z = 0.046$. The RMS noise is 3.1 μJy/beam. For MCXC J0352.4-7401 (Abell 3186), the total intensity 1.28 GHz MeerKAT images of this cluster shown in Fig. 5 have a resolution beam of 6.9″ × 7.1″. This corresponds to a spatial resolution of approximately 16 kpc at $z = 0.127$. The RMS noise is 2.6 μJy/beam. The resolution beam of the total intensity 1.28 GHz MeerKAT image of El Gordo shown in Fig. 6 is 7.1″ × 7.1″, corresponding to a physical resolution of approximately 55 kpc at $z = 0.87$. The RMS noise is 1.5 μJy/beam.

### Polarization measurements
We use the synchrotron polarization measurements obtained from JVLA observations[22,25]. We only report here the main characteristics of these observations while we refer to the original works for a detailed explanation of the data analysis. The magnetic-field orientation is defined as $\chi_B = \chi_0 + \pi/2$, inferred from the intrinsic polarization angle $\chi_0$ at the source obtained with the Rotation Measure synthesis technique[63]. This is intended to correct the measured Faraday rotation to represent the magnetic-field orientation at the relic.

Nevertheless, radio images can be affected by polarization leakage between Stokes parameters due to instrumental effects. The leakage from Stokes $I$ to $Q$ and $U$ was estimated at less than 2%[22,25]. Since relics are highly polarized, the leakage should have a marginal effect on the polarization angle estimates. The uncertainty of mapping magnetic fields with polarization can also arise from the adopted model of the distribution of thermal and relativistic electrons within the relics[64]. The uncertainty of the order 10° can serve as an estimate for the misalignment of polarization and SIG vectors, which corresponds to the reported AM variation of ± 0.2.

The magnetic-field orientation images of the RXC J1314.4-2515 galaxy cluster were obtained from a 1–2 GHz JVLA observation (with a central frequency of 1.5 GHz and resolution beam of 25″) while a 2–4 GHz JVLA observation (central frequency of 3 GHz and resolution beam of 30.5″) was used for Abell 2345. The polarization images are not smoothed further but re-gridded to match that of the SIG-measured magnetic-field orientation spatially. Polarization images are already masked to show only pixels detected with a corresponding Gaussian significance level greater than 5$\sigma$, as explained in refs. 22,25.

### Table 1 | Summary of global mean AM values in different Alfvén Mach number $M_A$ conditions

| $M_A$ | 2.4 | 2.9 | 3.2 | 5.2 | 7.8 |
|---|---|---|---|---|---|
| AM | 0.92 | 0.93 | 0.81 | 0.88 | 0.92 |

The uncertainty of AM calculated from the standard deviation of the mean is around 0.01–0.02. Source data are provided as a Source Data file.

**Table 2 | Summary of datasets used in this work**

| Cluster | Beam resolution (SIG) | $\langle\sigma_{\psi_g}\rangle$ | Frequency | Polarization data (resolution) |
|---|---|---|---|---|
| RXC J1314.4-2515[45] | 30 kpc (120 kpc) | 7.80° | 1.28 GHz | JVLA 3 GHz (120 kpc)[25] |
| Abell 2345[22] | 58 kpc (180 kpc) | E: 5.79°, W: 3.83° | 1.5 GHz | JVLA 1.5 GHz (110 kpc)[22] |
| Abell 3376[45] | 7 kpc (22 kpc) | E: 6.60°, W: 7.82° | 1.28 GHz | - |
| MCXC J0352.4-7401[45] | 16 kpc (62 kpc) | E: 7.44°, W: 7.52° | 1.28 GHz | – |
| El Gordo[45] | 55 kpc (400 kpc) | 7.41° | 1.28 GHz | – |

$\langle\sigma_{\psi_g}\rangle$ is the median value of the SIG's uncertainty over the region of interest. Source data are provided as a Source Data file.

## Data availability

The data are also accessible from the MeerKAT Galaxy Cluster Legacy Survey Data Release 1: https://archive-gw-1.kat.ac.za/public/repository/10.48479/7epd-w356/index.html and JVLA data archive: https://data.nrao.edu/portal/#/. The datasets generated and analyzed during the current study are available from the corresponding authors upon request. Source data are provided in this paper.

## Code availability

The simulations of MHD turbulence were conducted with the code MHDFlows, which can be downloaded from: https://github.com/MHDFlows/MHDFlows.jl. The analysis of the simulation data can be done with any science program language (e.g., Julia) according to the formulas provided explicitly in the manuscript. The SIG code supporting the findings of this study is available from the corresponding authors upon request. We have chosen not to publicly share the code in order to protect proprietary algorithms and maintain compliance with licensing agreements that restrict open distribution. We are committed to fostering scientific collaboration and are happy to provide the code to interested researchers for the purpose of replicating or building upon our findings.

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

## Acknowledgements

Y.H. and A.L. acknowledge the support of the NASA ATP AAH7546 and NSF grants AST 2307840. K.W.H. acknowledges the support of NASA ATP AAH7546 and the LDRD program of LANL with project # 20220107DR. NASA provided financial support for this work through award 09_0231 issued by the Universities Space Research Association, Inc. (USRA). This work used SDSC Expanse CPU at SDSC through allocations PHY230032, PHY230033, PHY230091, and PHY230105 from the Advanced Cyberinfrastructure Coordination Ecosystem: Services & Support (ACCESS) program, which is supported by National Science Foundation grants #2138259, #2138286, #2138307, #2137603, and #2138296. C.S. and A.B. acknowledge support from the MIUR grant FARE SMS and the ERC-StG DRANOEL, no. 714245

## Author contributions

All authors discussed the results, commented on the manuscript, and contributed to its writing. Y.H. and C.S. analyzed the observational data for radio emission and polarization, while Y.H. and K.W.H. conducted the numerical analysis. Y.H. and A.L. wrote the original manuscript, and A.B. and C.S. provided the observational data. G.B. provided crucial comments and suggestions on the application of SIG to ICM and on physical interpretations of the results.

## Competing interests

The authors declare no competing interests.
