## [Peer Review File · Nature Communications]

REVIEWER COMMENTS

Reviewer #1 (Remarks to the Author):

The paper "Synchrotron Intensity Gradient Revealing Magnetic Fields in Galaxy Clusters" applies a new methodology of mapping the magnetic field structure traced by gradients in the synchrotron intensity to galaxy cluster halos.

This is a unique approach, harboring the potential to better digest the structure of the magnetic fields in galaxy clusters, especially in the light of the new, observational perspective of SKA and SKA pathfinders. The novelty is that this method can be applied in the radio halo region of galaxy clusters, where depolarization is hiding the accessible magnetic field structure usually reflected in the polarized signal.

The results presented in the paper are very interesting and after considering some technical and scientific questions listed at the end of the report, the paper could be published in any, main astrophysical journal.

However, as the audience of nature communications is much more broad and not expert, there has to be spend some more, major effort to make the main article more understandable for non experts.

So in a nutshell (and sorry for some oversimplification), the authors want to demonstrate that observational one can for the first time confirm the picture, how we think magnetic field gets amplified when galaxy clusters merge. So basically the authors present (in figure 4 for example) an reconstruction of the observed, large scale magnetic field configuration which can be compared with the originally predictions in the seminal publication by Roettiger et al. 1999 (right column of figure 2). If robust, this can be seen as a milestone in observations of galaxy clusters magnetic field.

However, to make this point understandable for a more broader audience, the authors need to make some substantial changes on how the results are presented in addition to the technical/scientific questions.

--- structural comments ---

a) There needs to be a sketch (even artists like impression could be fine) reflecting the expected connection between merger geometry and magnetic field configuration, similar to how it nicely appears from the simulations in the figure from Roettiger et al. 1999.

b) In all figures, some (X-ray) contours reflecting the distribution of the ICM and at least the expected merger axis have to appear.

c) The main article starts to be very difficult to understand when substantial references to the supplementary material is made for arguments.

d) I think a little bit more discussion of the meaning of the magnetic field structure is needed. So I found it quite surprising that in the maps it appears as if there are even regions of 2Mpc size in linear direction which appear to have aligned magnetic fields across the radio halo, which should reflect a fully developed turbulence region.

--- technical and scientific questions ---

a) The main justification that the method works is the comparison with polarization data, observed for some of the relics. However, in the supplementary material, it is even discussed that at shocks the method will break down. And the relics are exactly these regions, where shocks are expected to be. Here, the agreement seem to be also best, where the magnetic field is aligned to the overall shape of the structure. In the point where you have complicated field structures, where the field is perpendicular to the structure, the two methods disagree. For example, in figure 2, upper panel. the intensity blob at -12.235/321.90. Also, it appears that wherever a bright spot is, there is a kind of dipole in the differences between the two methods. So in this sense, I see the danger that the new method gives statistically confirmation but just because it generally tends to be aligned with the larger scale structure. I think here, a better statistical prove has to be provided before claiming that the method works.

b) There should be a prove that the magnetic field structures seen in the observations indeed is related to what is seen in the MHD simulations. So one could compute structure functions for the reconstructed, observed magnetic field configuration in the halos compared with the MHD turbulence simulations.

c) When interpreting formulae (1) in the supplementary information, once should keep in mind that the 'literature' values quoted are mainly for the core, but the investigated radio halos (like in figure 4)

are extending into the cluster outskirts, where the values can be significantly different and therefore I_A might be somewhat different from the range given.

d) Regarding figure 6 and 7 in the supplementary material, especially when comparing to the merger axis for the radio halos, would be good to see how this would look like compared to random directions and also a test to define the merger axis based on the shape of the X-ray gas would be good to be checked.

e) Please add a length scale in lower panel of figure 1

f) Please change the limits of the color bars in figure 4 and 5 of the supplementary material. Having a red/blue color bar with only blue colors shown is quite useless.

Reviewer #2 (Remarks to the Author):

The manuscript contains the results of the synchrotron intensity gradient (SIG) study of the magnetic field in the Galaxy Clusters, Abell 2345, Abell 3376, MCXC J0352.4-7401 and El Gordo.

The article is well-written and logically developed. However, the uniqueness of the results may be questioned. The SIG technique has been reported in a number of literatures (Lazarian et al. 2017, 2018) and Hu et. al (2020).

SIG is useful for magnetic field direction / polarization measurements with acceptable agreement measures (AM). However, I have the following concerns;

Major comments

1. What is unique in the derived SIG with radio images compared to the SIG measurements reported by Lazarian et al. 2017? There also seem to be a generally lower AM from the results, please explain.
2. What percentage of the observationally derived polarization are attributed to instrumental polarization leakage? And how does this affect the derived AM?
3. It will be important to thoroughly discuss the regions within the studied objects in which the agreement measures are very low. What is the physical meaning of such deviation?

Minor comments

1. Figure 1. Acronym "AM" should be defined at or prior to first usage.
2. Figure 1 caption: left >> top left, right >> top right. Cite Knowles et al. (2022) at the reference to MeerKAT survey. Is there any reason for not doing the polarization study with the MeerKAT Galaxy Cluster Legacy Survey, MGCLS?
3. Figure 2: Provide reader with better perspective on Abell2345 by present the full image of the field alongside the zoom-ins.
4. Figure 3 caption:MeerKAT observation at 1.5GHz. The center frequency of MGCLS is 1.28 GHz, did the authors extrapolate the image to 1.5GHz?
5. Line 88, 89: "The bending of magnetic field...." This sentence is very confusing. Which magnetic field is bending? Jet magnetic field or the intracluster magnetic field? Chibueze et al. (2021) argued that the jets of the radio galaxy is bent by the intracluster magnetic field.
6. Figures 4 and 5 of Supplementary material: The colorscales can be adjusted to give a better impression of the gradients.

To all reviewers,

The reply from the authors is in blue if there are respective modifications they will be in red in the paper. The locations of the changes have been included respectively under each comment from referees. We thank the reviewers and editor for their numerous suggestions for improving the manuscript.

REVIEWER COMMENTS

Reviewer #1 (Remarks to the Author):

The paper "Synchrotron Intensity Gradient Revealing Magnetic Fields in Galaxy Clusters" applies a new methodology of mapping the magnetic field structure traced by gradients in the synchrotron intensity to galaxy cluster halos.

This is a unique approach, harboring the potential to better digest the structure of the magnetic fields in galaxy clusters, especially in the light of the new, observational perspective of SKA and SKA pathfinders. The novelty is that this method can be applied in the radio halo region of galaxy clusters, where depolarization is hiding the accessible magnetic field structure usually reflected in the polarized signal.

The results presented in the paper are very interesting and after considering some technical and scientific questions listed at the end of the report, the paper could be published in any, main astrophysical journal.

However, as the audience of nature communications is much more broad and not expert, there has to be spend some more, major effort to make the main article more understandable for non experts.

So in a nutshell (and sorry for some oversimplification), the authors want to demonstrate that observational one can for the first time confirm the picture, how we think magnetic field gets amplified when galaxy clusters merge. So basically the authors present (in figure 4 for example) an reconstruction of the observed, large scale magnetic field configuration which can be compared with the originally predictions in the seminal publication by Roettiger et al. 1999 (right column of figure 2). If robust, this can be seen as a milestone in observations of galaxy clusters magnetic field.

However, to make this point understandable for a more broader audience, the authors need to make some substantial changes on how the results are presented in addition to the technical/scientific questions.

We thank the referee for the above very helpful suggestions on how to make our results exciting to a broader audience. We edited the paper to emphasize that (1) The origin of magnetic fields in the large-scale structure is a fundamental part of cosmic magnetogenesis and is a field of major interest across different areas and communities. (2) The way magnetic fields are amplified in galaxy clusters and the connection between field properties and dynamics of galaxy clusters has been investigated in numerical simulations by different groups, eg., Roettiger (1999), Donnert, et al. (2018), Takizawa, et al., (2008), and Vazza et al. (2018). These simulations predict that magnetic fields evolve with cluster dynamics, **yet this point cannot be investigated through polarimetric studies due to their limitations.** (3) In this paper, we demonstrate that SIG can be used to investigate magnetic fields in galaxy clusters, establishing a new approach that is complementary to polarization studies and allowing us for the first time to carry out a first comparison between expectations from numerical simulations of merging clusters and observations. (4) Importantly, SIG opens the possibility to study the evolution of magnetic fields with cluster dynamics/evolution in the future, by applying SIG to large samples of clusters with different dynamics.

Following the referee's suggestion, we included a comparison of our findings with the predictions in the Roettiger et al. 1999 paper. We also pointed out the synergy of our approach to the polarization studies, in particular those that SKA and SKA pathfinder will make possible.

In particular, we wrote:

Line 14: "These magnetic fields are a fundamental part of cosmic magnetogenesis, either arising from the turbulent amplification of primordial seed fields or being injected by active galactic nuclei and galaxies. They are critical to maintaining energy balance within the intracluster medium (ICM) through heat conduction, coupling cosmic rays (CRs) to the intracluster gas, and generating synchrotron radiation through interactions with accelerated CR electrons. Despite the crucial importance, the origin of magnetic fields in ICM remains the grand challenge problem. To check the existing theoretical predictions that the magnetic field can be amplified during galaxy mergers (Roettiger 1999, Donnert, et al. 2018, Takizawa, et al., 2008, Vazza et al. 2018), a comprehensive understanding of magnetic field topology in galaxy clusters is imperative."

Line 117: "The reported magnetic field structure testifies the magnetic field amplification during galactic mergers. It is in line with previous RM grid measurements (Stuardi et al. 2021) and MHD simulations that predicted magnetic field evolve with cluster dynamics (Roettiger 1999, Donnert, et al. 2018, Takizawa, et al., 2008, Vazza et al. 2018, see Fig.~7 for an illustration.). The fields are stretched/stirred and further amplified by large-scale bulk flows along the merger axis."

Line 126: “Our results open new perspectives to map magnetic fields in clusters and large-scale structures and allow for the first comparison between numerical expectations of merging clusters and observations. ”

Line 130: “Being insensitive to the Faraday depolarization, SIG can be applied to many clusters with diffuse radio emission, which is especially timely in view of the coming SKA and the Low Frequency Array (LOFAR) observations. It opens a unique way of using radio data for the regions where depolarization masks and distorts the polarized signal.”

We realized that the audience of nature communications is much broader, so edited the manuscript in order to accommodate the referee’s suggestions.

--- structural comments ---

a) There needs to be a sketch (even artists like impression could be fine) reflecting the expected connection between merger geometry and magnetic field configuration, similar to how it nicely appears from the simulations in the figure from Roettiger et al. 1999.

We provided a sketch of the magnetic field evolution at different cluster merge stages, see the plot below.

Fig. 7. Cartoon illustration of magnetic field evolution in cluster merger. In the merge of two clusters (panel a), the magnetic field is draped and amplified at the merger (advancing) shock in the first phase (panel b), then the field is stretched along the merger axis (panel c), and finally is amplified by turbulence generated in the cluster (panel d).

b) In all figures, some (X-ray) contours reflecting the distribution of the ICM and at least the expected merger axis have to appear.

We edited the figures (see Figs. 1, 2, 4, 5, and 6) to include the X-ray contours and the expected merger axis.

c) The main article starts to be very difficult to understand when substantial references to the supplementary material is made for arguments.

We agree with the referee and thank him/her very much for the suggestion. We transferred the arguments to the main text. More details are left in the supplementary material so that an expert reader can test the rigor of our arguments. We hope that the paper is easier to read for the broad audience.

Please see Method 1. (1) Theoretical consideration, 1. (2) Comparison with polarization, 1. (5) Numerical testing

d) I think a little bit more discussion of the meaning of the magnetic field structure is needed. So I found it quite surprising that in the maps it appears as if there are even regions of 2Mpc size in linear direction which appear to have aligned magnetic fields across the radio halo, which should reflect a fully developed turbulence region.

The referee points out a very important effect. We expect that the observed structures arise due to the relative inefficiency of the turbulent dynamo at its non-linear stage. First of all, large-scale fields appear as a consequence of cluster merger, and turbulence is driven by the merger. However, less than 10% of cascading turbulent kinetic energy is

transferred to magnetic field energy (see Cho et al. 2009, Xu & Lazarian 2016). Thus, for stationary turbulence, this would take about 10 largest eddy turnover times for magnetic field energy to come to equipartition with kinetic turbulence motions. Merging clusters do not have so long time and therefore the more prominent magnetic field structure is developing through large-scale stretching determined by flows arising from the merger. We expect to see more magnetic field stochasticity on the small scale, but the picture that is observed with SIG is determined by the large-scale magnetic field structure that is mainly sampled via LOS averaging. This is the reason why the magnetic field structures appear parallel to the merging axis.

We added more discussion on this. In particular:

Line 120: “On the other hand, the presence of large-scale magnetic fields suggests that the magnetic field amplification through turbulent dynamo is rather inefficient at its non-linear stage, in which less than 10% of turbulent kinetic energy is transferred to magnetic field energy (Cho et al. 2009, Xu & Lazarian 2016). Thus for stationary turbulence, it would take around ten of the longest eddy turnover times for the magnetic field to reach equipartition with turbulence. Since the merging of clusters did not last for such a long time, the more prominent magnetic field structure is formed through the large-scale stretching caused by the flows resulting from the merger. We anticipate that there will be more magnetic field stochasticity on the small scale.”

--- technical and scientific questions ---

a) The main justification that the method works is the comparison with polarization data, observed for some of the relics. However, in the supplementary material, it is even discussed that at shocks the method will break down. And the relics are exactly these regions, where shocks are expected to be. Here, the agreement seem to be also best, where the magnetic field is aligned to the overall shape of the structure. In the point where you have complicated field structures, where the field is perpendicular to the structure, the two methods disagree. For example, in figure 2, upper panel. the intensity blob at -12.235/321.90. Also, it appears that wherever a bright spot is, there is a kind of dipole in the differences between the two methods. So in this sense, I see the danger that the new method gives statistically confirmation but just because it generally tends to be aligned with the larger scale structure. I think here, a better statistical prove has to be provided before claiming that the method works.

We apologize for not clarifying this important point in our original text and agree that our original presentation could confuse our reader on the role of shocks. We changed our presentation to explain why shocks are not important for our analysis.

First of all, SIG measures the field in the (spatially resolved) downstream region of shocks at 10-100 kpc distance from the shock front, where the bulk of the radio emission is generated. The deviation in the SIG direction from that of the underlying magnetic field happens for oblique shocks **only at the shock front**, due to the jump in intensity. The shock front is typically a very thin strip that occupies only one observational beam thickness along the shock front. The sub-block averaging (over a number of beams or pixels) method adopted in the SIG approach, on the other hand, marginalizes the contribution from a few beams along the shock front, provided that all other beams contain different information.

The elongated intensity structure in the figure below is a shock observed in our MHD simulations. Nevertheless, the numerical results below indicate, that the sharp intensity jump does not affect SIG (red segment) after the (square) sub-block averaging. We edited the text to clarify this point.

In general, differences between SIG and polarization are expected. First of all, SIG and polarization infer magnetic field directions using very different effects induced by magnetic fields along the LOS. SIG is a statistical technique based on the statistical properties of magnetized turbulent flows, while the polarization is sensitive to the value of the magnetic field strength perpendicular to the LOS and the density of relativistic electrons.

Second, there are uncertainties in both methods. Polarization is affected by depolarization effects and Faraday rotation, although we did our best to de-rotate the polarization angle and tried to use a high signal-to-noise ratio. For SIG, it is affected by noise in observation data, as we presented in the supplementary material. The disagreement typically appears in high uncertainty regions, see Fig. 2 and 3 in the supplementary. Thus we expect a global statistical agreement between the two.

As for the differences when a bright spot of emission is present in the total intensity, we attach here a zoom-in of the region together with a high-resolution radio image on the right:

At the position of the bright spot highlighted by the referee (magenta ellipse on the right), there is a point source. Since GT is only based on total synchrotron intensity, it is sensitive to the presence of this source while, if the source is unpolarized, it is not seen by the polarization method.

We added the following text:

Line 87: "We noticed that the misalignment is associated with a point source. These sources induce synchrotron intensity gradients that are not associated with magnetic

fields. The removal of the point source increases the correspondence between polarization and SIG increases. On the other hand, intensity jumps at shock fronts may induce deviation in the SIG from that of the underlying magnetic field. However, as we discuss in the Supplementary Information, the contribution of the shock fronts becomes marginal in the process of the sub-block averaging method adopted in SIG.”

In the supplementary information sec.2: “However, it is important to note that, in radio relics, SIG is actually measuring the magnetic field at the shock downstream regions, in which the jump of intensity does not appear, so SIG still works there. As for the shock front, it is typically a very thin strip that occupies only one observational beam. The sub-block averaging (over a number of beams) method adopted in SIG, on the other hand, diminishes the contribution from only one beam. Therefore, we expect that the contribution from turbulence or large-scale flows should dominate the gradient signal.”

b) There should be a prove that the magnetic field structures seen in the observations indeed is related to what is seen in the MHD simulations. So one could compute structure functions for the reconstructed, observed magnetic field configuration in the halos compared with the MHD turbulence simulations.

The SIG approach has been tested with an extensive set of 3D MHD numerical simulations that we provide in the Supplementary materials. An illustration of one of the synchrotron synthetic maps obtained with MHD simulations is presented below. We see that the 2D maps of the magnetic field obtained with SIG generally well agree with the synthetic polarization maps. The corresponding alignment measure characterizing the agreement AM is in the range of 0.81 - 0.93, which corresponds to the statistical deviation of the SIG-measured and polarization direction by 10 - 17 degrees. Similar to observations, we observe, some sub-blocks where the directions deviate. The nature of these deviations can be attributed to the stochastic nature of turbulence and the presence of the intermittent fast modes as discussed in Ho & Lazarian (2023). The structure functions of SIG and polarization are presented in supplementary Fig. 6.

The testing of SIG with cosmological MHD simulations of merging galaxy clusters with a self-consistent way of generating synchrotron radiation and sufficient resolution to reliably apply the SIG technique is on our agenda. Such simulations are not yet available.

In the text, we added the following:

Line 117: “It is in line with previous RM grid measurements and MHD simulations that predicted magnetic fields evolve with cluster dynamics (Donner et al . 2018, Roettiger et al. 1999, Takizawa et al. 2008, Vazza et al. 2018, see Fig. 7 for an illustration). Cluster mergers generate turbulent magnetic fields in the ICM. The fields are stretched/stirred and further amplified by large-scale bulk flows along the merger axis.”

Line 127: “SIG’s ability to trace the galaxy clusters’ magnetic fields is confirmed by (a) MHD simulations presented in the Method (see Fig. 8), (b) the correspondence of SIG with polarization measurements in radio relics, and (c) the correspondence of the SIG-traced halo magnetic field structure to those numerically predicted in the refs. 24, 46–48.”

c) When interpreting formulae (1) in the supplementary information, one should keep in mind that the 'literature' values quoted are mainly for the core, but the investigated radio halos (like in figure 4) are extending into the cluster outskirts, where the values can be significantly different and therefore I_A might be somewhat different from the range given.

We agree that I_A might vary, but this does not affect the applicability of the technique. In fact, we demonstrated in the Method that the technique works even for cases where I_A is not resolved. This is an important point suggested by the referee and we added a note to the text:

Line 144: “Note that quoted values of the physical quantities are typically measured for cluster cores. In the edges of the cluster, these values may be different.”

Line 149: “Therefore, the actual number of β_A is not important for using SIG to map the magnetic field, because the gradients are perpendicular to the magnetic field in both cases.”

d) Regarding figure 6 and 7 in the supplementary material, especially when comparing to the merger axis for the radio halos, would be good to see how this would look like compared to random directions and also a test to define the merger axis based on the shape of the X-ray gas would be good to be checked.

Thanks for the suggestions. We agree that the merger axis would be different when it is determined by X-ray or optical observations. We repeated the analysis by rotating the merger axis by 45 degrees and also include the cases that which the merger axis is determined by the X-ray emission's elongation. This would account for the uncertainty in the merger axis.

Supplementary Line 47: “To account for potential uncertainties, we repeat the analysis by rotating the merge axis by $\pm 45^\circ$. However, the merger axis can also be different when derived from the X-ray map elongation or the optical analysis of the merging sub-clusters. Here we further analyze the merger axis determined by the X-ray contour's elongation. The (X-ray) merger axis orients (north through east) $\sim 90^\circ$, $\sim -15^\circ$, $\sim 78^\circ$, $\sim -30^\circ$, $\sim -30^\circ$ for RXC J1314.4 - 2515, Abell 2345, Abell 3376, MCXC J0352.4 - 7401, and El Gordo, respectively.”

e) Please add a length scale in lower panel of figure 1

The scale length was added to the lower panel of Figure 1 as suggested by the referee. Thanks for the suggestion.

f) Please change the limits of the color bars in figure 4 and 5 of the supplementary material. Having a red/blue color bar with only blue colors shown is quite useless.

Thanks for the suggestions. We edited the figures and changed the colorbar in accordance with the referee's suggestions.

Reviewer #2 (Remarks to the Author):

The manuscript contains the results of the synchrotron intensity gradient (SIG) study of the magnetic field in the Galaxy Clusters, Abell 2345, Abell 3376, MCXC J0352.4-7401 and El Gordo.

The article is well-written and logically developed. However, the uniqueness of the results may be questioned. The SIG technique has been reported in a number of literatures (Lazarian et al. 2017, 2018) and Hu et. al (2020).

SIG is useful for magnetic field direction / polarization measurements with acceptable agreement measures (AM). However, I have the following concerns;

We thank the referee for his/her/their evaluation. The SIG technique was introduced by Lazarian et al. (2017, 2018) and tested by comparing it with the Planck polarization measurement of our **Milky Way**. In Hu et al. (2020), the SIG is applied to the **relaxed** cluster Perseus focusing on the very central region of the cluster, which indeed does not host a giant radio halo and radio relics extending on Mpc scales, as well as no available synchrotron polarization, exist. Both studies assumed that we could resolve the scale

λ_A of Alfvénic transition, which is true for the Milky Way studies but was not a default for the Perseus cluster. Thus the latter application of the SIG technique was questionable. As described in the corresponding paper.

The novelty of the paper in terms of the technique employed is that within the present study, we numerically demonstrated that the requirement of the SIG technique to resolve the scale λ_A is not required for the successful tracing of magnetic fields with SIG. This opens the possibility of using the technique for reliable studies of very weak magnetic fields, which is essential for studying magnetic fields in galaxy clusters and even at larger scales. Thus, in terms of applications of SIG, this paper is **the first reliable application** of SIG to galaxy clusters.

In terms of the new information, the novelty of this paper is as follows: (a) **the first application** of the SIG technique to **disturbed** galaxy clusters, covering the entire cluster volume or Mpc^3 regions; (b) **the first observational test** of the SIG performance in disturbed galaxy cluster using the **observed synchrotron polarization** in disturbed clusters and (c) **first magnetic field measurements in radio halos**, i.e. reveals the structure of magnetic fields on the largest scales ever measured.

We believe that the synergy of the above features makes this study unique and its results exciting to a broad audience. In addition, this paper opens an avenue for magnetic field studies at the scale of the Large Scale Universe structure, as the synchrotron intensity emission has been reported by LOFAR and MeerKAT (see Botteon et al. 2022, Cassano et al. 2023, Knowles, K. et al. 2022). We made these points clear while improving our paper.

Abstract: “Most importantly, the synchrotron intensity gradient is not limited by Faraday depolarization in the cluster central regions and allows us to map magnetic fields in the radio halos of RXC J1314.4 -2515 and El Gordo.”

Line 35: “Thus, so far, no polarization mapping of magnetic fields in radio halos has been carried out. This is one of the main challenges that will be faced by next-generation radio facilities, such as the Square Kilometre Array (SKA)”

Line 41: As a result, the direction of SIG determines the magnetic field, as was proven by comparing the magnetic fields traced by SIG and synchrotron polarization for the Milky Way magnetic fields (Lazarian et al 2017). This analysis was further substantiated through numerical simulations that replicated the turbulent conditions within the Milky Way (Lazarian et al 2017). Nevertheless, it is crucial to recognize that the nature of turbulence within galaxy clusters is distinctly different from the turbulence observed within the Milky Way (Li et al. 2020). Therefore, so far there has been only one attempt to apply SIG for magnetic field mapping in the relaxed cluster Perseus (Hu et al. 2020).

The application assumed the Alfvén scale, i.e., the scale at which magnetic fields become dynamically important (Brunetti & Lazarian 2007), is observationally resolved. Based on the current study of turbulence and magnetic field strength in the ICM, the Alfvén scale is constrained from 1 to 60 kpc (see the Method). However, our numerical simulations (see the Method) demonstrated that resolving the Alfvén scale is not essential for SIG. We showed that at scales larger than the Alfvén scale, magnetic fields passively follow the motion of the large-scale flow, with SIG aligned perpendicular to the magnetic field, similar to the case when the Alfvén scale is resolved (see Fig. 8 in the Method). This new understanding underlines our first confident application of SIG to the ICM.

Line 60: “...we present the first magnetic field measurements in RXC J1314.4-2515 and El Gordo radio halos, revealing the structure of magnetic fields on the largest scales ever measured.”

Line 126: “Our results open new perspectives to map magnetic fields in clusters and large-scale structures and allow for the first comparison between numerical expectations of merging clusters and observations”

Major comments:

1. What is unique in the derived SIG with radio images compared to the SIG measurements reported by Lazarian et al. 2017? There also seem to be a generally lower AM from the results, please explain.

Lazarian et al. (2017) introduced the SIG and compared it with the Planck polarization measurement of our Milky Way. The physical conditions in our galaxy (~hundreds pc scale) are very different from the galaxy clusters both in terms of the degree of magnetization (sub-Alfvénic or trans-Alfvénic versus super-Alfvénic) and scales (hundreds pc versus Mpc scale) used in this work. **In this paper, we present the first-ever measurement of the magnetic field structure in radio halos.**

In the Milky Way, the depolarization effect and Faraday rotation are less significant than those in galaxy clusters. Also, the signal-to-noise ratio in observation would be higher in our galaxy. Therefore, the uncertainties in both SIG and synchrotron polarization are expected to contribute to the lower AM (see also our answer for point 3 below).

Line 35: “Thus, so far, no polarization mapping of magnetic fields in radio halos has been carried out. This is one of the main challenges that will be faced by next-generation radio facilities, such as the Square Kilometre Array (SKA)”

Line 41: As a result, the direction of SIG determines the magnetic field, as was proven by comparing the magnetic fields traced by SIG and synchrotron polarization for the Milky Way magnetic fields (Lazarian et al 2017). This analysis was further substantiated through numerical simulations that replicated the turbulent conditions within the Milky Way (Lazarian et al 2017). Nevertheless, it's crucial to recognize that the nature of turbulence within galaxy clusters is distinctly different from the turbulence observed within the Milky Way (Li et al. 2020). Therefore, so far there has been only one attempt to apply SIG for magnetic field mapping in the relaxed cluster Perseus (Hu et al. 2020). The application assumed the Alfvén scale, i.e., the scale at which magnetic fields become dynamically important (Brunetti & Lazarian 2007), is observationally resolved. Based on the current study of turbulence and magnetic field strength in the ICM, the Alfvén scale is constrained from 1 to 60 kpc (see the Method). However, using numerical simulations (see the Method), the Alfvén scale is for the first time demonstrated not essential for SIG. At scales larger than the Alfvén scale, magnetic fields passively follow the motion of the large-scale flow, with SIG aligned perpendicular to the magnetic field, similar to the case when the Alfvén scale is resolved (see Fig. 8 in the Method). This new understanding encourages the first confident application of SIG to the ICM.

Line 60: “we present the first magnetic field measurements in RXC J1314.4-2515 and El Gordo radio halos, revealing the structure of magnetic fields on the largest scales ever measured.”

Line 78: “The measured AM is a bit lower compared to its AM obtained for SIG in Milky Way (Lazarian et al. 2017). We attribute this to the higher signal-to-noise and the smaller Faraday rotation effects in the Milky Way case.

Line 126: “Our results open new perspectives to map magnetic fields in clusters and large-scale structures and allow for the first comparison between numerical expectations of merging clusters and observations.”

2. What percentage of the observationally derived polarization are attributed to instrumental polarization leakage? And how does this affect the derived AM?

For JVLA data, the receiver-receiver leakage was corrected during the calibration procedure and the residual on-axis polarization leakage is <0.5%. The off-axis polarization leakage at the distance of the relic from the pointing center is below 2% (which is the maximum value reached by V/I , this is discussed in Stuardi et al. 2019 and 2021). Since radio relics are highly polarized (>10%) the leakage gives a marginal contribution to their polarization and thus to their polarization angle estimate. The uncertainty on the polarization angle derived from polarization analysis accounts for both the uncertainty on the polarized flux estimate and the one on the RM used to

de-rotate the angle. The average uncertainty on the polarization angle is 2.5 deg for RXCJ 1314, which is less than the average uncertainty of SIG for the same objects (i.e., 7.8 deg). The average uncertainty using polarization is 5.8 (4.8) deg for the eastern (western) radio relic in Abell 2345, while it is 5.8 (3.8) deg using SIG. However, we note that the largest uncertainty in the magnetic field direction derived from polarization comes from the assumption that the measured RM originates from an external Faraday screen. Within this approach, the volume distribution of emitting electrons, parallel to the line of sight and perpendicular components of the magnetic field and the thermal electrons is uncertain which can affect the uncertainty in the Faraday rotation effect (see Lazarian & Pogosyan 2016). The limitations of this approach for the particular cases of the two clusters are discussed in Stuardi et al. 2019 and 2021.

In terms of AM, the total uncertainty of ~ 10 deg from both polarization and SIG would give AM an uncertainty of ± 0.2 around.

Line 234: Radio images can be affected by polarization leakage between Stokes parameters due to instrumental effects. The leakage from Stokes I to Q and U was estimated at less than 2 % (Stuardi et al. 2019, 2020). Since relics are highly polarized, the leakage has a negligible effect on the polarization angle estimates. The uncertainty of identifying magnetic fields with polarization arises from the model of the distribution of thermal and relativistic electrons within the relics (Lazarian & Pogosyan 2016). The total uncertainty of $\sim 10^\circ$ is a reasonable estimate for the misalignment of polarization and SIG vectors, which corresponds to the reported AM variation of ± 0.2 .

3. It will be important to thoroughly discuss the regions within the studied objects in which the agreement measures are very low. What is the physical meaning of such deviation?

Actually, we do not expect a perfect agreement between SIG and polarization. It is important to stress that SIG and polarization employ different physical effects to reveal magnetic fields. SIG is based on MHD turbulence and traces the magnetic fields associated with large-scale flows and regions with larger local shear contribute more. At the same time, the polarization measure is sensitive to the value of the magnetic field strength perpendicular to the line of sight and the concentration of relativistic electrons. The direction of the magnetic field is changing in galaxy clusters along the line of sight. Thus the integration of the contributions from SIG and the polarization is not bound to coincide. A similar effect is well known while polarization from dust and synchrotron polarization are compared. We added the corresponding discussion to the paper (see Method 1. (2): Comparison with polarization).

The difference might also come from uncertainties in both methods. Polarization is affected by the depolarization effect and Faraday rotation, although we have re-rotated

the polarization angle and tried to use a high signal-to-noise ratio. For SIG, it is affected by noise in observation data, as we discussed in the supplementary material (Figs. 2 and 3).

Line 81: “In addition, unlike SIG, polarization is sensitive to Faraday rotation and Faraday depolarization. As a result, we do expect to see differences between the polarization and SIs. The systematic difference between the two measures carries important information that sheds light on the difference in the physical mechanisms of the processes that reveal magnetic field direction, and this difference can be explored in future studies to get deeper insight into the physics of ICM.”

Line 87: “In particular, we noticed that the misalignment is associated with a point source. These sources induce synchrotron intensity gradients that are not associated with magnetic fields. The removal of the point source increases the correspondence between polarization and SIG increases.”

Line 152: “SIG and synchrotron polarization are based on different physical effects to reveal the magnetic field. While synchrotron polarization emerges from the magnetic fields’ effects on relativistic electrons, SIG is grounded in the interaction between magnetic fields and conducting fluid. Notably, measurements obtained from both methods are subject to the effects of LOS averaging. Given the expected scenario where the LOS integration length for cluster halos surpasses the scale of magnetic field entanglement, turbulence-induced fluctuations are not anticipated to manifest a preferential direction, instead accumulating through a process akin to a random walk. These turbulence-driven fluctuations play a pivotal role in decreasing the synchrotron polarization fraction, known as Faraday depolarization. However, SIG is immune to the depolarization effect, maintaining its reliability in tracing the magnetic fields in super-Alfvenic radio halos. As illustrated in Fig. 8, the alignment between SIG and magnetic fields remains statistically stable, even when the LOS integration length increases. On the other side, polarization is sensitive not only to the value of the magnetic field strength that aligns parallel to the LOS but also to the density of thermal electrons within the environment. Given that the orientation of magnetic fields within galaxy clusters is subject to changes along the LOS, there is a possibility for differences between contributions to SIG and those to polarization.

Minor comments:

1. Figure 1. Acronym "AM" should be defined at or prior to first usage.

We edited the text to define “AM” at its first usage.

Line 66: “We define the Alignment Measure (AM) to quantify the alignment...”

2. Figure 1 caption: left >> top left, right >> top right. Cite Knowles et al. (2022) at the reference to MeerKAT survey. Is there any reason for not doing the polarization study with the MeerKAT Galaxy Cluster Legacy Survey, MGCLS?

We corrected the caption and provided the citation to Knowles et al. (2022). Also, thanks for drawing our attention to the MGCLS survey. The polarization data was not released when the paper was submitted. While polarization is now released for Abell 3376 and MCXC J0352.4 - 7401, it is still not processed to correct the Faraday rotation.

Nevertheless, we compared the SIG with available polarization for RXC J1314.4 -2515 and Abell 2345, as well as earlier polarization observation for Abell 3376 (see Kale et al. 2012, 2012MNRAS.426.1204K, and the left plot below of the Abell 3376's magnetic field extracted from Kale et al. 2012) and El Gordo (see Lindner et al. 2014, 2014ApJ...786...49L, and the left plot below of the El Gordo's magnetic field extracted from Lindner et al. 2014). We can see an agreement between the measurements.

Line 98: The SIG measurements are in agreement with earlier partial polarization observations in Abell 3376's double relics (Kale et al. 2012) and El Gordo's west relic (Lindner et al. 2014)

3. Figure 2: Provide reader with better perspective on Abell2345 by present the full image of the field alongside the zoom-ins.

We edited the figure and provided the full image of the cluster.

4. Figure 3 caption:MeerKAT observation at 1.5GHz. The center frequency of MGCLS is 1.28 GHz, did the authors extrapolate the image to 1.5GHz?

It should be 1.28 GHz. We corrected it.

Fig. 4, caption: "Top: overall synchrotron emission intensity map from MeerKAT observation at 1.28 GHz..."

5. Line 88, 89: "The bending of magnetic field..." This sentence is very confusing. Which magnetic field is bending? Jet magnetic field or the intracluster magnetic field? Chibueze et al. (2021) argued that the jets of the radio galaxy is bent by the intracluster magnetic field.

We meant the jet magnetic fields. The jets are affected by the ICM magnetic field, so the magnetic fields associated with the jets are also bent.

Line 99: The magnetic fields associated with radio galaxy jets and bent by ICM in Abell 3376 have also been observed in the SIG measurement (Chibueze et al. 2021).

6. Figures 4 and 5 of Supplementary material: The colorscales can be adjusted to give a better impression of the gradients.

Thanks for the suggestions. We edited the figures accordingly.

REVIEWERS' COMMENTS

Reviewer #1 (Remarks to the Author):

Thanks for the detailed treatment of my concerns/suggestions.

The manuscript can now be accepted as it is.

Reviewer #2 (Remarks to the Author):

The authors have addressed accordingly as the concerns raised, thus I recommend for publication.